# Boltzmann Weighting Done Right in Reinforcement Learning

## Abstract

The Boltzmann softmax operator can trade-off well between exploration and exploitation according to current estimation in an exponential weighting scheme, which is a promising way to address the exploration-exploitation dilemma in reinforcement learning. Unfortunately, the Boltzmann softmax operator is not a non-expansion, which may lead to unstable or even divergent learning behavior when used in estimating the value function. The convergence of value iteration is guaranteed in a restricted set of non-expansive operators and how to characterize the effect of such non-expansive operators in value iteration remains an open problem. In this paper, we propose a new technique to analyze the error bound of value iteration with the the Boltzmann softmax operator. We then propose the dynamic Boltzmann softmax(DBS) operator to enable the convergence to the optimal value function in value iteration. We also present convergence rate analysis of the algorithm. Using Q-learning as an application, we show that the DBS operator can be applied in a model-free reinforcement learning algorithm. Finally, we demonstrate the effectiveness of the DBS operator in a toy problem called GridWorld and a suite of Atari games. Experimental results show that outperforms DQN substantially in benchmark games.

## 1 Introduction

In sequential decision making problem, an agent learns to find an optimal policy that maximizes the expected discounted long-term reward, which can be modeled by a Markov decision process (MDP). In an MDP, the optimal value function is the fixed point of the Bellman operator. Thus, the optimal value function can be computed by iterative updates from arbitrary initial value function, i.e., value iteration (Bellman (2013)). Littman & Szepesvári (1996) proposed the generalized MDP considering generalized action selection operators, according to which the value function is estimated. If the generalized action selection operator satisfies the non-expansion property, the uniqueness of the fixed-point of the generalized Bellman operator is guaranteed, thus ensuring the convergence of generalized value iteration algorithm. Examples of non-expansive operators include the max and the mean operators.

Without full information about transition dynamics and reward function of the environment, in reinforcement learning, the agent aims to learn an optimal policy by interacting with the unknown environment from experience. Reinforcement learning has achieved groundbreaking success for many decision making problems, both in discrete and continuous domains, including robotics (Kober et al. (2013)), game playing (Mnih et al. (2015)), and many others. One of the most fundamental challenges is how to balance exploration and exploitation, where the exploration-exploitation dilemma occurs in action selection and value function optimization (Asadi & Littman (2016)). For value function optimization, the agent estimates the value function according to the action selection operator and then updates the estimated value. A number of effective algorithms to address exploration-exploitation dilemma have their root in alternating the action selection operator, which falls in the generalized MDP framework.

One of the most important reinforcement learning algorithm, Q-learning, employs the the max operator for value function optimization. The max operator always greedily selects the action that gives the best value and updates current estimation according to the value. As the current estimation for value functions is not accurate, such greedy selection may lead to misbehavior, e.g., the overesti-

mation phenomenon (Hasselt (2010)). In fact, the max operator is the extreme case of exploitation, lacking the ability to explore or consider other choices. On the other hand, the mean operator estimates the value by computing the average. Thus, the mean operator solely explores which fails to utilize current estimation.

The Boltzmann softmax operator is a natural summary operator that has been widely applied (Cesa-Bianchi et al. (2017)). Specifically, it is an exponentially weighting scheme, where the weights are computed according to the current estimation and its parameter $\beta$. The parameter $\beta$ trades off between exploration and exploitation. When $\beta \to \infty$, it behaves as the max operator which solely favors exploitation while it behaves as the mean operator when $\beta \to 0$ which purely focuses on exploration. However, despite from the advantages, it is very challenging to apply the operator in value function optimization. First, the parameter $\beta$ is difficult to choose (Sutton et al. (1998)). Second, as shown in (Littman & Szepesvári (1996); Asadi & Littman (2016)), the Boltzmann softmax operator is not a non-expansion, which may lead to multiple fixed-points and thus the value function of this policy is not well-defined. The non-expansive property is vital to guarantee the uniqueness of the fix-point and the convergence of the learning algorithm. Without such property, the learning algorithm may misbehave or even diverge. Thus, it is widely believed that the Boltzmann softmax operator cannot be used directly due to the violation of the non-expansive property (Littman & Szepesvári (1996); Asadi & Littman (2016)). In fact, as far as we know, how to characterize the use of the Boltzmann softmax operator, which violates the property of non-expansion in value iteration, remains an open problem (Littman (1996)).

In this paper, we study the property of the Boltzmann softmax operator and propose a new technique to characterize its error bound in value iteration with fixed parameter $\beta$. To be specific, we show that for the error bound between the value function induced by the Boltzmann softmax operator and the optimal value function, there remains a term related to $\beta$ that will not converge to $0$. Thus, although the Boltzmann softmax operator guarantees the approximate convergence of value iteration, it would converge to a sub-optimal policy unfortunately. Indeed, the direct use of the Boltzmann softmax operator inevitably introduces performance drop in value iteration.

We then take a step further and study an essential problem, *is there a way that the Boltzmann softmax operator be applied in value iteration which guarantees the convergence to the optimal policy?*

Based on this technique, we propose the dynamic Boltzmann softmax operator $\mathrm{boltz}_{\beta_t}$, termed the DBS operator, to eliminate the loss and enable the convergence of value iteration to the optimal value function. Our core idea is to dynamically change $\beta_t$ in value iteration and present its convergence rate analysis. Then, we propose the DBS Q -learning algorithm with the application of the DBS operator in a popular model-free reinforcement learning algorithm, i.e., Q-learning (Watkins & Dayan (1992)), and prove the convergence of the DBS Q-learning.

We conduct experiments to verify the effectiveness and efficiency of our proposed dynamic Boltzmann softmax operator. We first evaluate DBS value iteration and DBS Q-learning on a tabular case, the GridWorld. Results show that the DBS operator leads to smaller error and faster convergence. We then demonstrate that the DBS operator can be extended to large scale problems, Atari games. Using DQN as baseline, we show that DQN with the dynamic Boltzmann softmax operator (abbreviated as DBS-DQN) substantially outperforms DQN in a suite of Atari benchmark games.

## 2 PRELIMINARIES

A Markov decision process (MDP) is defined by a 5-tuple $(\mathcal{S}, \mathcal{A}, p, r, \gamma)$, where $\mathcal{S}$ and $\mathcal{A}$ denote the set of states and actions, $p(s'|s, a)$ represents the transition probability from state $s$ to state $s'$ under action $a$, and $r(s, a)$ is the corresponding immediate reward. The discount factor is denoted by $\gamma \in [0, 1)$, which controls the degree of importance of future rewards.

At each time, the agent interacts with the environment with its policy $\pi$, a mapping from state to action. The objective is to find an optimal policy that maximizes the expected discounted long-term reward $\mathbb{E}[\sum_{t=0}^{\infty} \gamma^t r_t | \pi]$, which can be solved by estimating value functions. The state value of $s$ and state-action value of $s$ and $a$ under policy $\pi$ are defined as $V^{\pi}(s) = \mathbb{E}_{\pi}[\sum_{t=0}^{\infty} \gamma^t r_t | s_0 = s]$ and $Q^{\pi}(s, a) = \mathbb{E}_{\pi}[\sum_{t=0}^{\infty} \gamma^t r_t | s_0 = s, a_0 = a]$. The optimal value functions are defined as $V^*(s) = \max_{\pi} V^{\pi}(s)$ and $Q^*(s, a) = \max_{\pi} Q^{\pi}(s, a)$.

Littman & Szepesvári (1996) proposed a general framework for reinforcement learning, where the generalized action selection operator is denoted by $\bigotimes$. The optimal value function $V^*$ satisfies the generalized Bellman equation, which is defined recursively as in Equation (1):

$$V^*(s) = \bigotimes_{a \in A} \left[ r(s, a) + \sum_{s' \in S} p(s'|s, a) \gamma V^*(s') \right] \tag{1}$$

Starting from arbitrary initial value function $V_0$, the optimal value function $V^*$ can be computed by value iteration (Bellman (2013)) according to an iterative update: $V_{k+1} = \mathcal{T} V_k$, where $\mathcal{T}$ is the generalized the Bellman operator as defined in Equation (2).

$$(\mathcal{T}V)(s) = \bigotimes_{a \in A} \left[ r(s, a) + \sum_{s' \in S} p(s'|s, a) \gamma V(s') \right]. \tag{2}$$

The convergence of value iteration is guaranteed if $\bigotimes$ is a non-expansion, which guarantees the unique solution of the generalized Bellman equation (1), i.e., $\mathcal{T}V^* = V^*$. The non-expansion is defined as:

$$\left| \bigotimes_a Q_1(s, a) - \bigotimes_a Q_2(s, a) \right| \leq ||Q_1(s, \cdot) - Q_2(s, \cdot)||_\infty, \tag{3}$$

where $|| \cdot ||_\infty$ denotes the $\ell_\infty$-norm.

The Boltzmann softmax operator is one kind of the action selection operator $\bigotimes$, which is defined as:

$$\text{boltz}_\beta(\mathbf{X}) = \frac{\sum_{i=1}^n x_i e^{\beta x_i}}{\sum_{i=1}^n e^{\beta x_i}}. \tag{4}$$

## 3 ANALYSIS OF THE BOLTZMANN SOFTMAX OPERATOR

In this section, we first analyze the property of the Boltzmann softmax operator and then propose a new technique to analyze its error bound in value iteration.

It has been shown that the Boltzmann softmax operator is not a non-expansion ((Littman & Szepesvári (1996); Asadi & Littman (2016))) as it does not satisfy Inequality (3). Indeed, the non-expansive property is vital to the convergence of the learning algorithm, which guarantees the uniqueness of the fixed point. We first analyze the property of the Boltzmann softmax operator, which paves the path for studying the effect of using such operators that violates the non-expansive property in value iteration.

**Proposition 1** *For $\beta > 0$ and $X, Y \in \mathbb{R}^n$, the Boltzmann softmax operator satisfies the following property:*

$$|boltz_\beta(\mathbf{X}) - boltz_\beta(\mathbf{Y})| \leq ||\mathbf{X} - \mathbf{Y}||_\infty + \frac{2 \log(n)}{\beta}, \tag{5}$$

*where $|| \cdot ||_\infty$ is the $\ell_\infty$-norm in $\mathbb{R}^n$.*

In Proposition 1, we show that although the Boltzmann softmax operator is not a non-expansive operator, the degree of the violation of the non-expansive property is controlled by $\beta$. The larger the value of $\beta$ is, the closer it is to the non-expansion. Due to space limit, we put the proof of Proposition 1 in Appendix A.

Next, we propose a new technique to characterize the error bound of value iteration with Boltzmann softmax operator in value iteration, where the full proof is referred to Appendix B.

**Theorem 1 (Error bound of value iteration with Boltzmann softmax operator)** *Let $V_t$ be the value function computed by the Boltzmann softmax operator at the $t$-th iteration and $V_0$ denote the initial value. After $t$ iterations,*

$$||V_t - V^*||_\infty \leq \gamma^t ||V_0 - V^*||_\infty + \frac{4 \log(|A|)(1 - \gamma^t)}{\beta(1 - \gamma)}. \tag{6}$$

Taking the limit of $t$ in both sides of Inequality (6), we obtain the following result:

**Corollary 1** *For the Boltzmann softmax operator* $\mathrm{boltz}_\beta$, *the error of value functions is* $\lim_{t \to \infty} ||V_{t+1} - V^*|| \leq \frac{4 \log(|A|)}{\beta(1-\gamma)}$.

Corollary 1 characterizes the error bound of value iteration with the Boltzmann softmax operator. With a fixed parameter $\beta$, the error is upper bounded by $\frac{4 \log(|A|)}{\beta(1-\gamma)}$, which decreases with an increasing value of $\beta$. Thus, the direct use of the Boltzmann softmax operator inevitably introduces performance drop in practice. Motivated by the theoretical findings, we propose the dynamic Boltzmann softmax operator, which enables the convergence to the optimal.

## 4 DYNAMIC BOLTZMANN SOFTMAX OPERATOR

In this section, we propose the dynamic Boltzmann softmax (DBS) operator $\mathrm{boltz}_{\beta_t}$ to eliminate the error in value iteration. Next, we give theoretical analysis of the proposed DBS value iteration algorithm. We prove that it converges to the optimal policy if $\beta_t$ approaches $\infty$, as shown in Theorem 2. We then present the convergence rate analysis in Theorem 3. Finally, we show that the DBS operator can be applied in a prominent model-free reinforcement learning algorithm, Q-learning, with convergence guarantee.

From Corollary 1, although the Boltzmann softmax operator can converge, it may suffer from error due to the violation of the non-expansive property. We propose the dynamic Boltzmann softmax (DBS) operator to eliminate the error, which is motivated by Corollary 1 that although $\mathrm{boltz}_\beta$ is not a non-expansive operator, it performs very close to the non-expansion when $\beta$ is large enough.

Based on the DBS operator, we design the corresponding DBS value iteration algorithm. DBS value iteration algorithm admits a dynamically changing series $\{\beta_t\}$ (line 1) and update the value function according to the dynamic Boltzmann softmax operator $\mathrm{boltz}_{\beta_t}$ (line 6). Thus, the way to update the value function is according to the exponential weighting scheme, which is related to both the current estimation value and the parameter $\beta_t$.

---

**Algorithm 1:** DBS Value Iteration

**Input:** An increasing series $\{\beta_t\}$; termination condition $\theta$
1  Initialize $V(s), \forall s \in \mathcal{S}$ arbitrarily
2  **for** *each episode* $t = 1, 2, ...$ **do**
3  $\quad$ $\Delta \leftarrow 0$
4  $\quad$ **for** *each* $s \in \mathcal{S}$ **do**
5  $\quad\quad$ $v \leftarrow V(s)$
6  $\quad\quad$ $V(s) \leftarrow \mathrm{boltz}_{\beta_t} \left( \sum_{s', r} p(s', r | s, a)[r + \gamma V(s')] \right)$
7  $\quad\quad$ $\Delta \leftarrow \max(\Delta, |v - V(s)|)$
8  $\quad$ **if** $\Delta < \theta$ **then**
9  $\quad\quad$ break

---

### 4.1 CONVERGENCE ANALYSIS

In Theorem 2, we demonstrate that the DBS operator can enable the convergence of DBS value iteration to the optimal. Due to space limit, see Appendix C for proof of the theorem.

**Theorem 2 (Convergence of value iteration with the DBS operator)** *For a sequence of dynamic Boltzmann softmax operator* $\{\beta_t\}$, *if* $\beta_t \to \infty$, $V_t$ *converges to* $V^*$, *where* $V_t$ *and* $V^*$ *denote the value function after* $t$ *iterations and the optimal value function.*

Theorem 2 implies that DBS value iteration does converge to the optimal policy if $\beta_t$ approaches infinity. Although the Boltzmann softmax operator may violate the non-expansive property for some values of $\beta$, we only need $\beta$ approaches infinity to guarantee the convergence.

The convergence rate of the DBS operator is shown in Theorem 3, where the proof is provided in Appendix C.

**Theorem 3 (Convergence rate of value iteration with the DBS operator)** *For any power series* $\beta_t = t^p (p > 0)$, *we have that for any* $\epsilon \in (0, \min\{0.25, ||R||^{-1}\})$, *after* $\max\{O\big(\frac{\log(\frac{1}{\epsilon}) + \log(\frac{1}{1-\gamma})}{\log(\frac{1}{\gamma})}\big), O\big((\frac{1}{(1-\gamma)\epsilon})^{\frac{1}{p}}\big)\}$ *steps, the error* $||V_t - V^*|| \leq \epsilon$.

For the larger value of $p$, the convergence rate is faster. Note that when $p$ approaches $\infty$, the convergence bound is dominated by the first term.

From the above theoretical analysis of the DBS operator, we demonstrate that value iteration can still converge to the optimal even with an operator violating the non-expansive property. Such finding generalizes previous understanding of the convergence of value iteration, which is restricted to a class of non-expansive operators. In addition, the convergence rate is of the same order as the standard Bellman operator. This implies that the DBS operator will not lose too much in terms of the convergence rate in value iteration. These findings provide theoretical background and pave the way for the study of the use of the DBS operator in reinforcement learning algorithms, e.g., Q-learning, which does not have full information about the model. In the absence of full information about the model, the agent has to explores in the environment and exploits the optimal strategy.

### 4.2 APPLICATION: Q-LEARNING

In this section, we show that the DBS operator can be applied in a model-free Q-learning algorithm (Algorithm 9), which requires careful trade-off between exploration and exploitation.

The DBS Q-learning updates the Q-value according to the DBS operator, where it admits a dynamically changing series $\beta_t$. It is worth noting that in Theorem 3, the larger value of $p$ results in faster convergence rate in value iteration. However, this is not the case in Q-learning. Indeed, Q-learning differs from value iteration in that it knows nothing about the environment, which means the agent has to learn from experience. Thus, the agent needs to balance between exploration and exploitation. If $p$ is too large, it quickly approximates the max operator that favors pure exploitation.

---

**Algorithm 2:** DBS Q-learning

**Input:** An increasing series $\{\beta_t\}$
1  Initialize $Q(s, a), \forall s \in \mathcal{S}, a \in \mathcal{A}$ arbitrarily, and $Q(terminal)\text{-}(state, \cdot) = 0$
2  **for** *each episode* $t = 1, 2, ...$ **do**
3  $\quad$ Initialize $s$
4  $\quad$ $\beta_t = f(t)$
5  $\quad$ **for** *each step of episode* **do**
6  $\quad\quad$ choose $a$ from $s$ using policy derived from $Q$
7  $\quad\quad$ take action $a$, observe $r, s'$
8  $\quad\quad$ $Q(s, a) \leftarrow Q(s, a) + \alpha[r + \gamma boltz_{\beta_t}(Q(s, \cdot)) - Q(s, a)]$
9  $\quad\quad$ $s \leftarrow a'$

---

In Theorem 4, we prove that DBS Q-learning converges to the optimal policy under the same additional condition as in DBS value iteration. The proof is based on the stochastic approximation lemma in (Singh et al. (2000)), and the full proof is referred to Appendix E.

**Theorem 4 (Convergence of DBS Q-learning)** *The Q-learning algorithm with dynamic Boltzmann softmax policy given by*

$$Q_{t+1}(s_t, a_t) = (1 - \alpha_t(s_t, a_t))Q_t(s_t, a_t) + \alpha_t(s_t, a_t)[r_t + \gamma boltz_{\beta_t}(Q_t(s_{t+1}, \cdot))] \quad (7)$$

*converges to the optimal* $Q^*(s, a)$ *values if*

1. *The state and action spaces are finite.*

2. $\sum_t \alpha_t(s, a) = \infty$ *and* $\sum_t \alpha_t^2(s, a) < \infty$

3. $\lim_{t \to \infty} \beta_t = \infty$

4. *Var$(r(s, a))$ is bounded.*

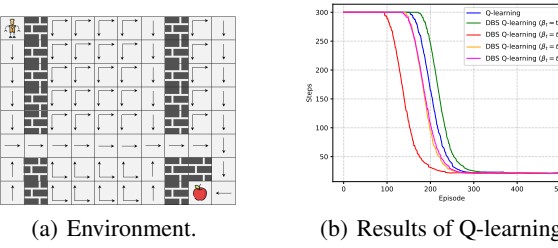

(a) Environment.                    (b) Results of Q-learning.

Figure 1: The grid world experiment.

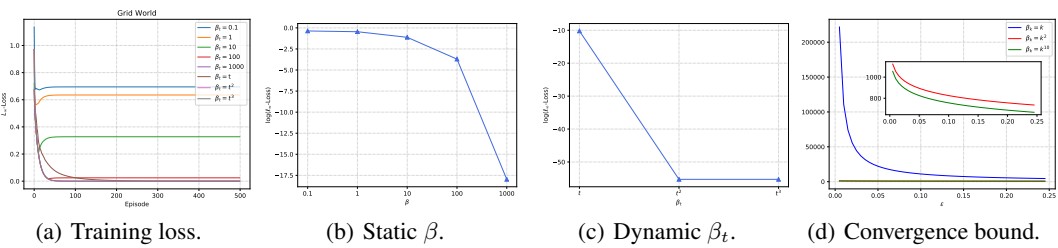

(a) Training loss.          (b) Static $\beta$.          (c) Dynamic $\beta_t$.          (d) Convergence bound.

Figure 2: Value iteration in the GridWorld.

Note that different from value iteration, Q-learning does not know the full information about the model and has to learn from experience. Thus, it is vital to trade-off exploration and exploitation. Unlike the max operator, the DBS operator enables exploration in the begining of learning with a small value of $\beta_t$. Since the estimated value function is not accurate in the begining of learning, it is better to weight possible choices according to the current estimation rather than greedily selecting the maximum estimated value. The DBS favors exploitation more as $\beta_t$ increases, meaning that it is able to utilize the information of current estimation.

## 5 EXPERIMENTS

### 5.1 GRIDWORLD

We first evaluate the performace of DBS value iteration DBS Q-learning in a toy problem, the GridWorld (Figure 1(a)), which is a larger variant of the environment of (O'Donoghue et al. (2016)). The GridWorld consists of $10 \times 10$ grids, with the dark grids representing walls. The agent starts at the upper left corner and aims to eat the apple at the bottom right corner upon receiving a reward of $+1$. Otherwise, the reward is 0. An episode ends if the agent successfully eats the apple or a maximum number of steps 300 is reached. For this experiment, we consider the discount factor $\gamma = 0.9$.

The training loss of value iteration is shown in Figure 2(a). As expected, larger value of $\beta$ leads to smaller loss. Figure 2(b) and Figure 2(c) demonstrate the training loss in logarithmic form for the last episode. For static $\beta$, the value iteration algorithm suffers from some loss which decreases as $\beta$ increases. For dynamic $\beta_t$, the performance of $t^2$ and $t^3$ are the same and achieve the smallest loss. The convergence rate is illustrated in Figure 2(d). For higher order $p$ of $\beta_t = t^p$, the convergence rate is faster. We also see that the convergence rate of $t^2$ and $t^{10}$ is very close as discussed before.

Figure 1(b) demonstrates the number of steps the agent spent in each episode. DBS Q-learning with $\beta_t = t^2$ achieves the best performance as it best trades off between exploration and exploitation. When the power $p$ of $\beta_t = t^p$ increases, it performs closer to the max operator for exploitation. When $p = 1$, it performs worse than Q-learning in this simple game as it explores more. Thus, considering trading off between exploration and exploitation, we choose $p = 2$ in the following experiments.

| | Mean | Median |
|---|---|---|
| **DQN** | 495.76% | 84.72% |
| **DBS-DQN** | 1611.49% | 103.95% |

Table 1: Summary of Atari games.

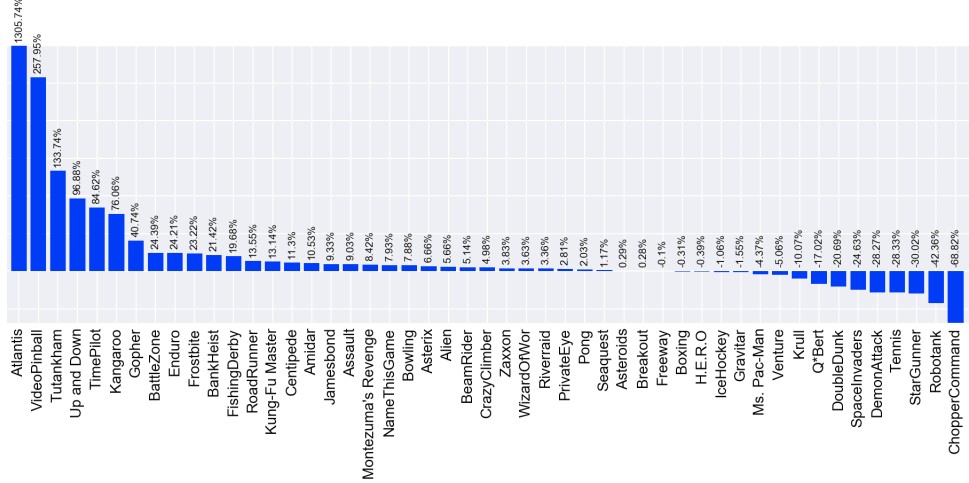

Figure 3: Relative human normalized score on Atari games.

## 5.2 ATARI

We evaluate the DBS-DQN algorithm on 49 Atari games from the Arcade Learning Environment (Bellemare et al. (2013)) by comparing it with DQN. For fair comparison, we use the same setup of network architectures and hyper-parameters as in Mnih et al. (2015) for both DQN and DBS-DQN. Note that DBS-DQN estimates the value for the next state according to the DBS operator, where $\beta_t = ct^2$ and $c$ is the coefficient. See Appendix F for full implementation details. For each game, we train each algorithm for 50M steps. The evaluation procedure is identical to Mnih et al. (2015), 30 no-op evaluation, where the agent performs a random number (up to 30) of "do nothing" actions in the beginning of an episode.

Table 1 shows the summary of results in human normalized score, which is defined as (Van Hasselt et al. (2016)):

$$\frac{\text{score}_{\text{agent}} - \text{score}_{\text{random}}}{\text{score}_{\text{human}} - \text{score}_{\text{random}}} \times 100\%, \qquad (8)$$

where human score and random score are taken from Wang et al. (2015). As illustrated in Table 1, DBS-DQN significantly outperforms DQN in terms of both the mean and the median of the human normalized score. To better characterize the effectiveness of DBS-DQN, its improvement over DQN is shown in Figure 3, where the improvement is defined as the relative human normalized score:

$$\frac{\text{score}_{\text{agent}} - \text{score}_{\text{baseline}}}{\max\{\text{score}_{\text{human}}, \text{score}_{\text{baseline}}\} - \text{score}_{\text{random}}} \times 100\%, \qquad (9)$$

with DQN serving as the baseline. In all, DBS-DQN exceeds the performace of DQN in 33 out of 49 Atari games. Full scores of comparison is referred to Appendix G. Figure 4 shows the learning curves for each algorithm. The results provide empirical evidence that the DBS operator trades-off well exploration and exploitation in value function optimization.

## 6 RELATED WORK

In reinforcement learning, avoiding the exploration-exploitation dilemma is a vital task. The Boltzmann softmax operator is a popular way to balance exploration and exploitation by exponentially

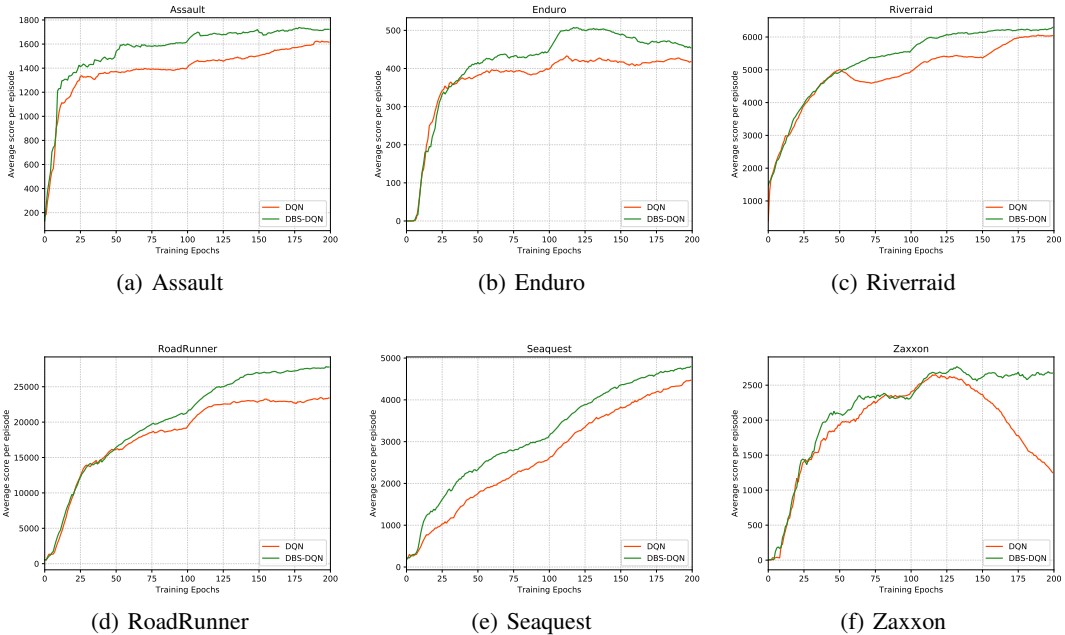

Figure 4: Learning curves in Atari games.

weighting its current estimation (Kaelbling et al. (1996)). To address the problem, a line of research focuses on the exploration strategy where the agent should exploit current best action on the one hand, but it needs to explore whether there are better possibilities on the other hand. (Singh et al. (2000)) studied the convergence of on-policy algorithm, i.e., Sarsa. They show that the strategy needs to be greedy in the limit, which guarantees that the optimal action can be selected. Although they considered a dynamic parameter of $\beta$ in the Boltzmann softmax operator, it depends on the state, which is impractical in complex problems as Atari games. The other line studies the use of alternative operators in value function estimation. To better trade-off between exploration and exploitation, Asadi & Littman (2016) proposed the "Mellowmax" operator, where the degree of exploration and exploitation is controlled by its parameter. However, although the "Mellowmax" operator can approximate maximization in the limit, it converges to a sub-optimal policy rather than the optimal policy. Haarnoja et al. (2017) utilized the log-sum-exp operator, which enables better exploration and learns deep energy-based policies.

It is worth noting that meta learning cannot be applied to solve the problem since the Boltzmann softmax operator with a fixed parameter would inevitably lead to error in value iteration, so there is no optimal value of $\beta$.

## 7 CONCLUSION

We provide a new theoretical technique to analyze the error bound of the value iteration algorithm with the Boltzmann softmax operator. Then, we develop the DBS value iteration algorithm based on our proposed dynamic Boltzmann softmax (DBS) operator which enables convergence to the optimal value function and present convergence rate analysis. We show that the DBS operator can be applied in a model-free reinforcement learning algorithm, Q-learning. Experimental results demonstrate the effectiveness of the DBS operator and show that it can be extended to complex problems as Atari games. For future work, it is worth studying the sample complexity of our proposed DBS Q-learning algorithm. It is also a promising direction to apply the DBS operator to other state-of-the-art DQN-based algorithms, such as Rainbow (Hessel et al. (2017)).

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

## A  PROPERTY OF THE BOLTZMANN SOFTMAX OPERATOR

**Proposition 1** *For $\beta > 0$ and $X, Y \in \mathbb{R}^n$, the Boltzmann softmax operator satisfies the following property:*

$$|boltz_\beta(\mathbf{X}) - boltz_\beta(\mathbf{Y})| \leq ||\mathbf{X} - \mathbf{Y}||_\infty + \frac{2\log(n)}{\beta}, \tag{10}$$

*where $\|\cdot\|_\infty$ is the $\ell_\infty$-norm in $\mathbb{R}^n$.*

**Proof 1** *Let $L_\beta$ denote a log-sum-exp function, i.e., $L_\beta(\mathbf{X}) = \frac{1}{\beta}\log(\sum_{i=1}^n e^{\beta x_i})$*

$$|boltz_\beta(\mathbf{X}) - boltz_\beta(\mathbf{Y})| = |(boltz_\beta(\mathbf{X}) - L_\beta(\mathbf{X})) - (boltz_\beta(\mathbf{Y}) - L_\beta(\mathbf{Y})) + (L_\beta(\mathbf{X}) - L_\beta(\mathbf{Y}))|$$
$$\leq |boltz_\beta(\mathbf{X}) - L_\beta(\mathbf{X})| + |boltz_\beta(\mathbf{Y}) - L_\beta(\mathbf{Y})| + |L_\beta(\mathbf{X}) - L_\beta(\mathbf{Y})| \tag{11}$$

*From MacKay & Mac Kay (2003), we have $|boltz_\beta(\mathbf{X}) - L_\beta(\mathbf{X})| \leq \frac{\log(|\mathbf{X}|)}{\beta}$*

*Substitute the above inequality into Equation (11), we have*

$$|boltz_\beta(\mathbf{X}) - boltz_\beta(\mathbf{Y})| \leq \frac{2\log(n)}{\beta} + |\frac{1}{\beta}\log(\sum_{i=1}^n e^{\beta x_i}) - \frac{1}{\beta}\log(\sum_{i=1}^n e^{\beta y_i})|$$
$$= \frac{2\log(n)}{\beta} + \frac{1}{\beta}|\log(\frac{\sum_{i=1}^n e^{\beta x_i}}{\sum_{i=1}^n e^{\beta y_i}})| \tag{12}$$

*Let $\Delta_i = |x_i - y_i|$ and assume $\sum_{i=1}^n e^{\beta x_i} \geq \sum_{i=1}^n e^{\beta y_i}$ without loss of generality. Then, we have*

$$\frac{1}{\beta}|\log(\frac{\sum_{i=1}^n e^{\beta x_i}}{\sum_{i=1}^n e^{\beta y_i}})| \leq \frac{1}{\beta}|\log(\frac{\sum_{i=1}^n e^{\beta(y_i+\Delta_i)}}{\sum_{i=1}^n e^{\beta y_i}})|$$
$$\leq \frac{1}{\beta}|\log(\frac{\sum_{i=1}^n e^{\beta(y_i+||\Delta_i||_\infty)}}{\sum_{i=1}^n e^{\beta y_i}})| \tag{13}$$
$$\leq ||\Delta_i||_\infty$$
$$= ||x - y||_\infty$$

*So we have*

$$|boltz_\beta(\mathbf{X}) - boltz_\beta(\mathbf{Y})| \leq \frac{2\log(n)}{\beta} + ||x - y||_\infty \tag{14}$$

## B  ERROR BOUND OF THE BOLTZMANN SOFTMAX OPERATOR

**Theorem 1 (Error bound of value iteration with Boltzmann softmax operator)** *Let $V_t$ be the value function computed by the Boltzmann softmax operator at the $t$-th iteration and $V_0$ denote the initial value. After $t$ iterations, $||V_t - V^*||_\infty \leq \gamma^t ||V_0 - V^*||_\infty + \frac{4\log(|A|)(1-\gamma^t)}{\beta(1-\gamma)}$.*

**Proof 2** *Let $\mathcal{T}_\beta$ and $\mathcal{T}_m$ denote the dynamic programming operators for the Boltzmann softmax operator $boltz_\beta$ and the max operator respectively, so*

$$(\mathcal{T}_\beta V)(s) = boltz_\beta(Q(s, \cdot)), \; (\mathcal{T}_m V)(s) = \max(Q(s, \cdot)) \tag{15}$$

*Thus,*

$$||(\mathcal{T}_\beta V_1) - (\mathcal{T}_m V_2)||_\infty \leq \underbrace{||(\mathcal{T}_\beta V_1) - (\mathcal{T}_\beta V_2)||_\infty}_{(A)} + \underbrace{||(\mathcal{T}_\beta V_2) - (\mathcal{T}_m V_2)||_\infty}_{(B)} \tag{16}$$

*For the term $(A)$, we have*

$$||(\mathcal{T}_\beta V_1) - (\mathcal{T}_{\beta_t} V_2)||_\infty = \max_s |boltz_\beta(Q_1(s, \cdot)) - boltz_\beta(Q_2(s, \cdot))|$$

$$\leq \max_s \max_a |Q_1 - Q_2| + \frac{2\log(|A|)}{\beta}$$

$$\leq \max_s \max_a \gamma \sum_{s'} p(s'|s, a)|V_1(s') - V_2(s')| + \frac{2\log(|A|)}{\beta} \qquad (17)$$

$$\leq \gamma||V_1 - V_2||_\infty + \frac{2\log(|A|)}{\beta}$$

*For the term $(B)$, we have*

$$||(\mathcal{T}_\beta V_1) - (\mathcal{T}_m V_1)||_\infty = \max_s |boltz_\beta(Q_1(s, \cdot)) - \max(Q_1(s, \cdot))|$$

$$\leq \max_s \left[ |boltz_\beta(Q_1(s, \cdot)) - L_\beta(Q_1(s, \cdot))| + |L_\beta(Q_1(s, \cdot)) - \max(Q_1(s, \cdot))| \right]$$

$$\leq \frac{2\log(|A|)}{\beta}$$

$$(18)$$

*Thus,*

$$||(\mathcal{T}_\beta V_1) - (\mathcal{T}_m V_2)||_\infty \leq \gamma||V_1 - V_2||_\infty + \frac{4\log(|A|)}{\beta} \qquad (19)$$

*As for the max operator, $\mathcal{T}_m$ is a contraction mapping, then from Banach fixed-point theorem we have $\mathcal{T}_m V^* = V^*$*

*Let $\mathcal{T}_b^t$ denote the dynamic programming operator for a sequence of Boltzmann softmax operators, then we have*

$$||V_t - V^*||_\infty = ||\mathcal{T}_b^t V_0 - \mathcal{T}_m^t V^*||_\infty$$

$$\leq \gamma||(\mathcal{T}_b^{t-1} V_0 - \mathcal{T}_m^{t-1} V^*||_\infty + \frac{4\log(|A|)}{\beta}$$

$$\leq ...$$

$$\leq \gamma^t||V_0 - V^*||_\infty + \frac{4\log(|A|)}{\beta} \sum_{k=1}^t \gamma^{t-k} \qquad (20)$$

$$= \gamma^t||V_0 - V^*||_\infty + \frac{4\log(|A|)(1 - \gamma^t)}{\beta(1 - \gamma)}$$

## C  CONVERGENCE OF DBS VALUE ITERATION

**Theorem 2 (Convergence of value iteration with the DBS operator)**  *For a sequence of dynamic Boltzmann softmax operator $\{\beta_t\}$, if $\beta_t \to \infty$, $V_t$ converges to $V^*$, where $V_t$ and $V^*$ denote the value function after $t$ iterations and the optimal value function.*

**Proof 3** *Let $\mathcal{T}_{\beta_t}$ and $\mathcal{T}_m$ denote the dynamic programming operators for dynamic Boltzmann softmax operator $boltz_{\beta_t}$ and the max operator respectively, so $(\mathcal{T}_{\beta_t} V)(s) = boltz_{\beta_t}(Q(s, \cdot))$, $(\mathcal{T}_m V)(s) = \max(Q(s, \cdot))$. Thus, we have*

$$||(\mathcal{T}_{\beta_t} V_1) - (\mathcal{T}_m V_2)||_\infty \leq \underbrace{||(\mathcal{T}_{\beta_t} V_1) - (\mathcal{T}_{\beta_t} V_2)||_\infty}_{(A)} + \underbrace{||(\mathcal{T}_{\beta_t} V_2) - (\mathcal{T}_m V_2)||_\infty}_{(B)} \qquad (21)$$

*By similar techniques as in Theorem 6, we have*

$$||(\mathcal{T}_{\beta_t} V_1) - (\mathcal{T}_m V_2)||_\infty \leq \gamma||V_1 - V_2||_\infty + \frac{4\log(|A|)}{\beta_t} \qquad (22)$$

As the $\max$ operator $T_m$ is a contraction mapping, then from Banach fixed-point theorem we have $\mathcal{T}_m V^* = V^*$

Let $\mathcal{T}_b^t$ denote the dynamic programming operator for a sequence of dynamic Boltzmann softmax operators, so $\mathcal{T}_b^t = \mathcal{T}_{\beta_t} \mathcal{T}_{\beta_{t-1}} ... \mathcal{T}_{\beta_1}$, then we have

$$
\begin{aligned}
||V_t - V^*||_\infty &= ||\mathcal{T}_b^t V_0 - \mathcal{T}_m^t V^*||_\infty \\
&= ||(\mathcal{T}_{\beta_t}...\mathcal{T}_{\beta_1})V_0 - (\mathcal{T}_m...\mathcal{T}_m)V^*||_\infty \\
&\leq \gamma ||(\mathcal{T}_{\beta_{t-1}}...\mathcal{T}_{\beta_1})V_0 - (\mathcal{T}_m...\mathcal{T}_m)V^*||_\infty + \frac{4\log(|A|)}{\beta_t} \\
&\leq ... \\
&\leq \gamma^t ||V_0 - V^*||_\infty + 4\log(|A|) \sum_{k=1}^{t} \frac{\gamma^{t-k}}{\beta_k}
\end{aligned}
\tag{23}
$$

Since $\lim_{k\to\infty} \frac{1}{\beta_k} = 0$, we have that $\forall \epsilon_1 > 0, \exists K > 0,$ such that $\forall k > K, |\frac{1}{\beta_k}| < \epsilon_1$. Thus,

$$
\begin{aligned}
\sum_{k=1}^{t} \frac{\gamma^{t-k}}{\beta_k} &= \sum_{k=1}^{K(\epsilon_1)} \frac{\gamma^{t-k}}{\beta_k} + \sum_{k=K(\epsilon_1)+1}^{t} \frac{\gamma^{t-k}}{\beta_k} \\
&\leq \frac{1}{\min\beta} \sum_{k=1}^{K(\epsilon_1)} \gamma^{t-k} + \epsilon_1 \sum_{k=K(\epsilon_1)+1}^{t} \gamma^{t-k} \\
&= \frac{1}{\min\beta} \frac{\gamma^{t-K(\epsilon_1)}(1-\gamma^{K(\epsilon_1)})}{1-\gamma} + \epsilon_1 \frac{1(1-\gamma^{t-K(\epsilon_1)})}{1-\gamma} \\
&\leq \frac{1}{1-\gamma} \left( \frac{\gamma^{t-K(\epsilon_1)}}{\min\beta} + \epsilon_1 \right)
\end{aligned}
\tag{24}
$$

If $t > \frac{\log((\epsilon_2(1-\gamma)-\epsilon_1)\min\beta)}{\log\gamma} + K(\epsilon_1)$ and $\epsilon_1 < \epsilon_2(1-\gamma)$, then $\sum_{k=1}^{t} \frac{\gamma^{t-k}}{\beta_k} < \epsilon_2$.

So we obtain that $\forall \epsilon_2 > 0, \exists T > 0,$ such that $\forall t > T, |\sum_{k=1}^{t} \frac{\gamma^{t-k}}{\beta_k}| < \epsilon_2$.

Thus, $\lim_{t\to\infty} \sum_{k=1}^{t} \frac{\gamma^{t-k}}{\beta_k} = 0$.

Taking the limit of Equation (23), we have that

$$
\lim_{t\to\infty} ||V_{t+1} - V^*||_\infty \leq \lim_{t\to\infty} \left[ \gamma^t ||V_1 - V^*||_\infty + 4\log(|A|) \sum_{k=1}^{t} \frac{\gamma^{t-k}}{\beta_k} \right] = 0
\tag{25}
$$

## D   CONVERGENCE BOUND

**Theorem 3 (Convergence rate of value iteration with the DBS operator)** *For any power series* $\beta_t = t^p (p > 0)$*, we have that for any* $\epsilon \in (0, \min\{0.25, ||R||^{-1}\})$*, after* $\max\{O\left(\frac{\log(\frac{1}{\epsilon})+\log(\frac{1}{1-\gamma})}{\log(\frac{1}{\gamma})}\right), O\left((\frac{1}{(1-\gamma)\epsilon})^{\frac{1}{p}}\right)\}$ *steps, the error* $||V_t - V^*|| \leq \epsilon$.

**Proof 4**

$$
\begin{aligned}
\sum_{k=1}^{t} \frac{\gamma^{t-k}}{k^p} &= \gamma^t \left[ \sum_{k=1}^{\infty} \frac{\gamma^{-1}}{k^p} - \sum_{k=t+1}^{\infty} \frac{\gamma^{-1}}{k^p} \right] \\
&= \gamma^t \big[ \underbrace{\mathrm{Li}_p(\gamma^{-1})}_{\text{Polylogarithm}} - \gamma^{-(t+1)} \underbrace{\Phi(\gamma^{-1}, p, t+1)}_{\text{Lerch transcendent}} \big]
\end{aligned}
\tag{26}
$$

By Ferreira & López (2004), we have

$$
Eqaution(26) \approx \gamma^t \frac{\gamma^{-(t+1)}}{\gamma^{-1}-1} \frac{1}{(t+1)^p} = \frac{1}{(1-\gamma)(t+1)^p}
\tag{27}
$$

*From Theorem 2 we have*

$$||V_t - V^*|| \leq \gamma^t ||V_1 - V^*|| + \frac{4 \log(|A|)}{(1-\gamma)(t+1)^p} \leq 2 \max\{\gamma^t ||V_1 - V^*||, \frac{4 \log(|A|)}{(1-\gamma)(t+1)^p}\} \quad (28)$$

*Thus, for any $\epsilon > 0$, after at most $t = \max\{\frac{\log(\frac{1}{\epsilon})+\log(\frac{1}{1-\gamma})+\log(||R||)+\log(4)}{\log(\frac{1}{\gamma})}, \left(\frac{8 \log(|A|)}{(1-\gamma)\epsilon}\right)^{\frac{1}{p}} - 1\}$
steps, we have $||V_t - V^*|| \leq \epsilon$.*

## E    CONVERGENCE OF DBS Q-LEARNING

**Theorem 4 (Convergence of DBS Q-learning)** *The Q-learning algorithm with dynamic Boltzmann softmax policy given by*

$$Q_{t+1}(s_t, a_t) = (1 - \alpha_t(s_t, a_t))Q_t(s_t, a_t) + \alpha_t(s_t, a_t)[r_t + \gamma boltz_{\beta_t}(Q_t(s_{t+1}, \cdot))] \quad (29)$$

*converges to the optimal $Q^*(s, a)$ values if*

1. *The state and action spaces are finite.*
2. *$\sum_t \alpha_t(s, a) = \infty$ and $\sum_t \alpha_t^2(s, a) < \infty$*
3. *$\lim_{t \to \infty} \beta_t = \infty$*
4. *$Var(r(s, a))$ is bounded.*

**Proof 5** *Let $\Delta_t(s, a) = Q_t(s, a) - Q^*(s, a)$ and $F_t(s, a) = r_t + \gamma boltz_{\beta_t}(Q_t(s_{t+1}, \cdot)) - Q^*(s, a)$*

*Thus, from (29) we have $\Delta_{t+1}(s, a) = (1 - \alpha_t(s, a))\Delta_t(s, a) + \alpha_t(s, a)F_t(s, a)$, which has the same form as the process defined in Lemma 2.*

*Next, we verify $F_t(s, a)$ meets the required properties.*

$$\begin{aligned}
F_t(s, a) &= r_t + \gamma boltz_{\beta_t}(Q_t(s_{t+1}, \cdot)) - Q^*(s, a) \\
&= (r_t + \gamma \max_{a+1} Q_t(s_{t+1}, a_{t+1}) - Q^*(s, a)) + \gamma(boltz_{\beta_t}(Q_t(s_{t+1}, \cdot)) - \max_{a_{t+1}} Q_t(s_{t+1}, a_{t+1})) \\
&\triangleq G_t(s, a) + H_t(s, a)
\end{aligned}$$
$$(30)$$

*For $G_t$, it is indeed the $F_t$ function as that in Q-learning with static exploration parameters, which satisfies*

$$||\mathbb{E}[G_t(s, a)]|P_t||_w \leq \gamma ||\Delta_t||_w \quad (31)$$

*For $H_t$, we have*

$$\begin{aligned}
|\mathbb{E}[H_t(s, a)]| &= \gamma \big| \sum_{s'} p(s'|s, a)[boltz_{\beta_t}(Q_t(s', \cdot)) - \max_{a'} Q_t(s', a')] \big| \\
&\leq \gamma \big| \max_{s'}[boltz_{\beta_t}(Q_t(s', \cdot)) - \max_{a'} Q_t(s', a')] \big| \\
&\leq \gamma \max_{s'} \big| boltz_{\beta_t}(Q_t(s', \cdot)) - \max_{a'} Q_t(s', a') \big| \\
&= \gamma \max_{s'} \big| (boltz_{\beta_t}(Q_t(s', \cdot)) - L_{\beta_t}(Q_t(s', \cdot))) + (L_{\beta_t}(Q_t(s', \cdot)) - \max_{a'} Q_t(s', a')) \big| \\
&\leq \gamma \max_{s'} \big( |boltz_{\beta_t}(Q_t(s', \cdot)) - L_{\beta_t}(Q_t(s', \cdot))| + |L_{\beta_t}(Q_t(s', \cdot)) - \max_{a'} Q_t(s', a')| \big) \\
&\leq \frac{\gamma \log(|A|)}{\beta_t} + \gamma \max_{s'} \big| \frac{1}{\beta_t} \log(\sum_{a' \in A} e^{\beta_t Q_t(s', a')}) - \max_{a'} Q_t(s', a') \big| \\
&\leq \frac{2\gamma \log(|A|)}{\beta_t}
\end{aligned}$$
$$(32)$$

*Let $h_t = \frac{2\gamma \log(|A|)}{\beta_t}$, so we have*

$$||\mathbb{E}[F_t(s, a)]|P_t||_w \leq \gamma ||\Delta_t||_w + h_t, \quad (33)$$

*where $h_t$ converges to 0*

# F   IMPLEMENTATION DETAILS

The network architecture is the same as in (Mnih et al. (2015)). The input to the network is a raw pixel image, which is pre-processed into a size of $84 \times 84 \times 4$. Table 2 summarizes the network architecture.

| LAYER | TYPE | CONFIGURATION | ACTIVATION FUNCTION |
|---|---|---|---|
| 1st | convolutional | #filters=32, size=$8 \times 8$, stride=4 | ReLU |
| 2nd | convolutional | #filters=64, size=$4 \times 4$, stride=2 | ReLU |
| 3rd | convolutional | #filters=64, size=$3 \times 3$, stride=1 | ReLU |
| 4th | fully-connected | #units=512 | ReLU |
| output | fully-connected | #units=#actions | — |

Table 2: Network architecture.

# G    ATARI SCORES

| GAMES | DQN | DBS-DQN |
|---|---|---|
| Alien | 20.18 | **25.84** |
| Amidar | 56.73 | **67.26** |
| Assault | 780.99 | **851.52** |
| Asterix | 50.03 | **56.69** |
| Asteroids | 1.38 | **1.68** |
| Atlantis | 1651.23 | **23211.93** |
| Bank Heist | 59.66 | **81.08** |
| Battle Zone | 79.08 | **103.46** |
| Beam Rider | 49.89 | **55.04** |
| Bowling | 19.84 | **27.72** |
| Boxing | **732.5** | 730.25 |
| Breakout | 1332.64 | **1336.32** |
| Centipede | 25.86 | **37.16** |
| Chopper Command | 80.81 | **12.0** |
| Crazy Climber | 399.15 | **419.03** |
| Demon Attack | **659.59** | 473.09 |
| Double Dunk | 545.45 | **432.58** |
| Enduro | 84.72 | **108.93** |
| Fishing Derby | 163.77 | **196.0** |
| Freeway | **104.05** | 103.95 |
| Frostbite | 17.15 | **40.37** |
| Gopher | 395.37 | **556.44** |
| Gravitar | **9.44** | 7.89 |
| H.E.R.O. | **65.14** | 64.75 |
| Ice Hockey | **76.86** | 75.8 |
| James Bond | 270.09 | **295.29** |
| Kangaroo | 241.6 | **425.36** |
| Krull | **639.28** | 574.89 |
| Kung-Fu Master | 114.78 | **129.87** |
| Montezumas Revenge | 0.0 | **8.42** |
| Ms. Pac-Man | **41.81** | 37.45 |
| Name This Game | 102.76 | **110.91** |
| Pong | 113.88 | **116.19** |
| Private Eye | 0.18 | **2.98** |
| Q*Bert | **97.46** | 80.44 |
| River Raid | 38.27 | **41.63** |
| Road Runner | 504.66 | **573.03** |
| Robotank | **636.08** | 409.02 |
| Seaquest | 13.8 | 14.97 |
| Space Invaders | **101.55** | 76.54 |
| Star Gunner | **559.34** | 391.4 |
| Tennis | **232.26** | 166.45 |
| Time Pilot | 78.38 | **163.0** |
| Tutankham | 36.3 | **170.04** |
| Up and Down | 84.74 | 181.62 |
| Venture | **13.73** | 8.67 |
| Video Pinball | 12792.59 | **45791.36** |
| Wizard Of Wor | 51.05 | **54.68** |
| Zaxxon | 58.32 | **62.15** |

Figure 5: Human normalized scores across all games, starting with 30 no-op actions.

