# OpenReview forum: "A Convergent Variant of the Boltzmann Softmax Operator in Reinforcement Learning"
_ICLR.cc/2019/Conference_

### Official Review · AnonReviewer2 · 2018-11-01
**Boltzmann Weighting Done Right in Reinforcement Learning**

**Rating:** 5
**Confidence:** 4

**Review:**

I liked this paper overall, though I feel that the way it is pitched to the reader is misguided. The looseness with which this paper uses 'exploration-exploitation tradeoff' is worrying. This paper does not attack that tradeoff at all really, since the tradeoff in RL concerns exploitation of understood knowledge vs deep-directed exploration, rather than just annealing between the max action and the mean over all actions (which does not incorporate any notion of uncertainty). Though I do recognize that the field overall is loose in this respect,  I do think this paper needs to rewrite its claims significantly. In fact it can be shown that Boltzmann exploration that incorporates a particular annealing schedule (but no notion of uncertainty) can be forced to suffer essentially linear regret even in the simple bandit case (O(T^(1-eps)) for any eps > 0) which of course means that it doesn't explore efficiently at all (see Singh 2000, Cesa-Bianchi 2017). Theorem 4 does not imply efficient exploration, since it requires very strong conditions on the alphas, and note that the same proof applies to vanilla Q-learning, which we know does not explore well.

I presume the title of this paper is a homage to the recent 'Boltzmann Exploration Done Right' paper, however, though the paper is cited, it is not discussed at all. That paper proved a strong regret bound for Boltzmann-like exploration in the bandit case, which this paper actually does not for the RL case, so in some sense the homage is misplaced. Another recent paper that actually does prove a regret bound for a Boltzmann policy for RL is 'Variational Bayesian Reinforcement Learning with Regret Bounds', which also anneals the temperature, this should be mentioned.

All this is not to say that the paper is without merit, just that the main claims about exploration are not valid and consequently it needs to be repositioned. If the authors do that then I can revise my review.

Algorithm 2 has two typos related to s' and a'.

---

> ### Author Response · Authors · 2018-11-18
> **To Reviewer2**
>
> Thank you very much for the thoughtful reviews, especially for the exploration-exploitation trade-off.
>
> In this paper, we aim to make the Boltzmann softmax operator converge from the view of trade-off between exploration and exploitation in value function optimization, instead of the traditional understanding in the action selection process. To be specific, in stochastic environments, the max operator updates the value estimator in a ‘hard’ way by greedily summarizing action-value functions according to current estimation. However, this may not be accurate due to noise in the environment. Even in deterministic environments, this may not be correct either. This is because the estimate for the value is not correct in the early phase of the learning process. We elaborate this and distinguish it from the exploration-exploitation trade-off in the updated version in Section 2.2 and Section 5.1.
>
> Considering the title would be misleading, we change it accordingly.
>
> Thank you for pointing out the reference paper. We cite and discuss the paper in the updated version in Section 6 (Related Work).

---

### Official Review · AnonReviewer1 · 2018-11-02
**I don't think the theoretical results represent a significant advance**

**Rating:** 4
**Confidence:** 4

**Review:**

The writing and organization of the paper are clear.  Theorem 1 seems fine but is straightforward to anyone who has studied this topic and knows the literature.  Corollary one may be technically wrong (or at least it doesn't follow from the theorem), though this can be fixed by replacing the lim with a limsup.  Theorem 4 seems to be the main result all the work is leading up to, but I think this is wrong.  Stronger conditions are required on the sequence \beta_t, along the lines discussed in the paragraph on Boltzmann exploration in Section 2.2 of Singh et al 2000.  The proof provided by the authors relies on a "Lemma 2" which I can't find in the paper.  The computational results are potentially interesting but call for further scrutiny.  Given the issues with the theoretical results, I think its hard to justify accepting the paper.

---

> ### Author Response · Authors · 2018-11-18
> **Clarification to Reviewer1**
>
> Thank you for the comments. We are afraid that you have some misunderstandings for our work.
>
> Q1: Theorem 1 is straightforward.
> A1: The effect of operators which are not non-expansion when applied in value iteration is an open problem and worth studying (Algorithms for Sequential Decision Making, Littman, 1996). Although error bounds of value iteration with the traditional max operator is well-established, there’s no results for the Boltzmann softmax operator which violates the property of non-expansion.
>
> In Theorem 1, we propose a novel analysis to characterize the error bound of the Boltzmann operator when applied in value iteration. Please note that this is the first time that the analysis is presented, and it is of vital importance as value iteration is the basis for RL algorithms.
>
> Q2: Corollary 1 may be technically wrong.
> A2: Please note that ||·||_{\infty} denotes the L-\infty norm, and ||V_0 - V^*||_{\infty}, \log{|A|}, \beta, and \gamma are all constants which will not change by taking the limit of t. Corollary 1 is derived by taking the limit of t in both sides of Inequality (6) in Theorem 1.
>
> Q3: Theorem 4 may be wrong. Stronger conditions are required.
> A3: Theorem 4 is correct. In our DBS Q-learning algorithm, the action selection policy is epsilon-greedy. Thus, states will be visited infinitely often. In addition, different from (Singh et al., 2000), where they study on-policy reinforcement learning algorithm (Sarsa), \beta is state-independent here and thus is more flexible. Please also note that the main result of the paper is the characterization of the (dynamic) Boltzmann softmax operator in value iteration (Theorem 1, Theorem 2, and Theorem 3). We then apply the DBS operator in a well-known off-policy reinforcement learning algorithm, i.e Q-learning, and Theorem 4 is to guarantee the convergence of the resulting DBS Q-learning algorithm.
>
> Q4: Cannot find Lemma 2.
> A4: Lemma 2 refers to the stochastic approximation lemma (Lemma 1) in Section 3.1 of (Singh et al., 2000).

---

> > ### Comment · AnonReviewer1 · 2018-11-23
> > **response to clarifications**
> >
> > Reviewer 1 is right that corollary 1 is ok as is.
> >
> > Where in Section 4.2 does it say that actions are selected to be epsilon-greedy.  If that is the case, with fixed epsilon, Theorem 4 will be correct.  But I don't see where that is assumed.  Further, if that is assumed, its a poor choice of exploration scheme.
> >
> > I still can't verify the proof of Theorem 4.

---

> > > ### Author Response · Authors · 2018-12-16
> > > **Response**
> > >
> > > Theorem 4 applies to any action selection policies that guarantees infinite visitations for states and actions, and epsilon-greedy is an example policy that satisfy the requirement. Please note that a common choice for such policy is epsilon greedy (e.g. the DQN algorithm). Although epsilon varies, it decays from 1.0 to 0.1 and remains 0.1 thereafter. As epsilon is not 0, it still guarantees infinite visits for states.

---

### Official Review · AnonReviewer3 · 2018-11-03
**Okay paper but relatively thin novelty**

**Rating:** 4
**Confidence:** 5

**Review:**

Summary: This work demonstrates that, although the Boltzmann softmax operator is not a non-expansion, a proposed dynamic Boltzmann operator (DBS) can be used in conjunction with value iteration and Q-learning to achieve convergence to V* and Q*, respectively. This time-varying operator replaces the traditional max operator. The authors show empirical performance gains of DBS+Q-learning over Q-learning in a gridworld and DBS+DQN over DQN on Atari games.

Novelty: (1) The error bound of value iteration with the Boltzmann softmax operator and convergence & convergence rate results in this setting seem novel. (2) The novelty of the dynamic Boltzmann operator is somewhat thin, as (Singh et al. 2000) show that a dynamic weighting of the Boltzmann operator achieves convergence to the optimal value function in SARSA(0). In that work, the weighting is state-dependent, so the main algorithmic novelty in this paper is removing the dependence on state visitation for the beta parameter by making it solely dependent on time. A question for the authors: How does the proof in this work relate to / differ from the convergence proofs in (Singh et al. 2000)?

Clarity: In the DBS Q-learning algorithm, it is unclear under which policy actions are selected, e.g. using epsilon-greedy/epsilon-Boltzmann versus using the Boltzmann distribution applied to the Q(s, a) values. If the Boltzmann distribution is used then the algorithm that is presented is in fact expected SARSA and not Q-learning. The paper would benefit from making this clear.

Soundness: (1) The proof of Theorem 4 implicitly assumes that all states are visited infinitely often, which is not necessarily true with the given algorithm (if the policy used to select actions is the Boltzmann policy). (2) The proof of Theorem 1 uses the fact that |L(Q) - max(Q)| <= log(|A|) / beta, which is not immediately clear from the result cited in McKay (2003). (3) The paper claims in the introduction that “the non-expansive property is vital to guarantee … the convergence of the learning algorithm.” This is not necessarily the case -- see Bellemare et al., Increasing the Action Gap: New Operators for Reinforcement Learning, 2016.

Quality: (1) I appreciate that the authors evaluated their method on the suite of 49 Atari games. This said, the increase in median performance is relatively small, the delta being about half that of the increase due to double DQN. The improvement in mean score in great part stems from a large improvement occurs on Atlantis.

There are also a number of experimental details that are missing. Is the only change from DQN the change in update rule, while keeping the epsilon-greedy rule? In this case, I find a disconnect between the stated goal (to trade off exploration and exploitation) and the results. Why would we expect the Boltzmann softmax to work better when combined to epsilon-greedy? If not, can you give more details e.g. how beta was annealed over time, etc.?

Finally, can you briefly compare your algorithm to the temperature scheduling method described in Fox et al., Taming the Noise in Reinforcement Learning via Soft Updates, 2016?

Additional Comments:
(1) It would be helpful to have Atari results provided in raw game scores in addition to the human-normalized scores (Figure 5). (2) The human normalized scores listed in Figure 5 for DQN are different than the ones listed in the Double DQN paper (Van Hasselt et al, 2016). (3) For the DBS-DQN algorithm, the authors set beta_t = ct^2 - how is the value of c determined? (4) Text in legends and axes of Figure 1 and Figure 2 plots is very small. (5) Typo: citation for MacKay - Information Theory, Inference and Learning Algorithms - author name listed twice.

Similarly, if the main contribution is DBS, it would be interesting to have a more in-depth empirical analysis of the method -- how does performance (in Atari or otherwise) vary with the temperature schedule, how exploration is affected, etc.?

After reading the other reviews and responses, I still think the paper needs further improvement before it can published.

---

> ### Author Response · Authors · 2018-11-18
> **To Reviewer3**
>
> Thank you for the comments. Please find our responses as below, especially for the novelty of the work.
>
> Q1: The novelty of the DBS operator.
> A1: First of all, thank you for viewing our analysis for DBS novel. As we mentioned in the paper and showed by the corresponding title, we mainly aim to enable the convergence of the widely-used Boltzmann operator by a better exploration-exploitation trade-off, which is dispensable for reinforcement learning. As far as we know, it is the first time that we find a variant of the Boltzmann operator with good convergence rate.
>
> Although the state-dependent weighting of Boltzmann operator is proposed in (Singh et al. 2000), our DBS operator is state-independent and can scale to high-dimensional state space, which is crucial for RL algorithms. Furthermore, their operator is for on-policy RL algorithm, i.e. SARSA, while our DBS is for value iteration (a basic algorithm to solve the MDP) and Q-learning (a more popular off-policy RL algorithm). Therefore, our Q-learning algorithm with DBS is novel.
>
> Due to the difference of our algorithms and that in (Singh et al. 2000), we develop new techniques to prove the convergence. Specifically, for value iteration, we propose a novel analysis to characterize the error bound of value iteration with the Boltzmann operator, prove the convergence and present convergence rate analysis; for Q-learning, we leverage the stochastic approximation lemma (SA Lemma) presented in (Singh et al. 2000), which is an extension of the classic stochastic approximation theorem proven in (Jaakkola et al. 1994), to relate the process to the well-defined stochastic process in SA Lemma and then we quantify the additional term using similar techniques in our Theorem 1. Our results of value iteration have little relation with (Singh et al. 2000) and are mainly based on our own analysis (Proposition 1, Theorem 1, Theorem 2, and Theorem).
>
> Q2: What is the action selection policy? The states should be visited infinitely.
> A2: In our DBS Q-learning algorithm, the action selection policy is epsilon-greedy. Thus, states will be visited infinitely often. We make it clearer in the updated version.
>
> Please note that, the exploration-exploitation dilemma here is related to value function optimization (Asadi et al. 2017), rather than the traditional view of exploring the environment and exploiting the action during the action selection process. In stochastic environments, the max operator updates the value estimator in a ‘hard’ way by greedily summarizing action-value functions according to current estimation. However, this may not be accurate due to noise in the environment. Even in deterministic environments, this may not be accurate either. This is because the estimate for the value is not correct in the early stage of the learning process. We elaborate the effect of exploration and added empirical study in the updated version, please refer to Section 5.1.
>
> Q3: |L(Q) - max(Q)| <= log(A||) / beta is not immediately clear.
> A3: We give more details of the proof in the updated version, please refer to Appendix B.
>
> Q4: Non-expansion is not necessary for convergence.
> A4: Yes, non-expansion is an important and widely-used sufficient condition to guarantee the convergence of the learning problem (Littman 1996, Asadi et al. 2017). In this understanding, we say non-expansion is ‘vital’ for convergence. (Bellemare et al. 2016) proposed an alternative sufficient condition different from the non-expansion property. However, the condition is still not enough to cover common operators violating non-expansion such as the Boltzmann softmax operator.
>
> Q5: Detailed comments for the experiment.
> A5: Here are our quick feedbacks.
> 1) We compare with G-learning and analyze the effect in the updated version (Section 5.1).
> 2) We change the score to raw game scores in the updated version (Appendix H).
> 3) Please note that our score listed is exactly the same with ‘Dueling Network Architectures for Deep Reinforcement Learning’ and ‘Rainbow’, where the (original) scores for DQN are raw scores.
> 4) In our experiments, c is in [0, 1], and we have tuned the value of c in some of the games. This is because different games have different features and should have different values of c.
> 5) We have redrawn the plots to make it more reader-friendly and corrected some typos in the updated version.

---

> > ### Public Comment · (anonymous) · 2018-12-11
> > **Questions regarding the proofs**
> >
> >
> > 1. For Q3 above, there is no discussion in Appendix B regarding the bound of |L(Q) - max(Q)| (At least in the current version)
> >
> > 2. For the bound of |boltz(Q) - L(Q)|, could you please point out the corresponding page number in MacKay's book?

---

> > > ### Author Response · Authors · 2018-12-16
> > > **Response**
> > >
> > > Thanks a lot for your reply. We have actually updated the paper accordingly, however, it seems that due to system errors our paper is not the latest version. We are sorry about this.
> > >
> > > A1:
> > > We derive the bound of |L_b(X) - max(X)| ≤ log(n)/b by showing that max(X) ≤ L_b(X) ≤ max(X) + log(n)/b as follows:
> > > As b*max(X) = log(e^(max(b*X))) ≤ log(∑e^(b*x_i)), we have b*max(X) ≤ b*L_b(X)
> > > As b*max(X) + log(n) = log(e^(max(b*X))) + log(n) = log(max(e^(b*X))) + log(n) = log(n*max(e^(b*X))) ≥ log(∑e^(b*x_i)), we have b*L_b(X) ≤ b*max(X) + log(n)
> > > Combining these inequalities, we have max(X) ≤ L_b(X) ≤ max(X) + log(n)/b.
> > >
> > > A2:
> > > Please refer to Exercise 31.1 (a) of page 402 in Mackay’s book (https://www.ece.uvic.ca/~agullive/Mackay.pdf). To the best of our knowledge, there is no proof of the bound of | L_b(X) - boltz_b(X) |, and we give a proposition here:
> > >
> > > L_b (X) - boltz_b(X) = 1/b ∑-p_i log(p_i), where p_i is the weight of the Boltzmann distribution, i.e. p_i = e^(b*x_i)/∑e^(b*x_j). The proof is as follows:
> > > 1/b ∑-p_i log(p_i) = 1/b ∑ ( -e^(b*x_i)/∑e^(b*x_j) ) * log( e^(b*x_i)/∑e^(b*x_j) )
> > >                                = 1/b ∑ ( -e^(b*x_i)/∑e^(b*x_j) ) * ( b*x_i - log( ∑e^(b*x_j) ) ) )
> > >                                = -∑ ( ( e^(b*x_i) * x_i ) / ∑e^(b*x_j) ) + 1/b * log( ∑e^(b*x_j) )
> > >                                = -boltz_b(X) + L_b(X)
> > >
> > > As L_b(X) ≥ boltz_b(X), we have | L_b(X) - boltz_b(X) | = 1/b ∑-p_i log(p_i), where the right hand side is equal to the entropy of the Boltzmann distribution. The maximum of the right hand side is achieved when p_i=1/n, and equals to log(n)/b.
> > > Thus, we have | L_b(X) - boltz_b(X) | ≤ log(n)/b.

---

### Author Response · Authors · 2018-11-18
**Summary of the updated version**

We thank the reviewers for their careful reading and thoughtful reviews. We have updated the submission accordingly and the main changes in the updated version of the paper include:

+ we elaborate more about the exploration-exploitation dilemma in value function optimization
+ we add empirical analysis of the exploration-exploitation dilemma
+ we compare with G-learning in the GridWorld
+ we discuss more related papers
+ we refine experimental results of Atari
+ we elaborate the details for the proof of Theorem 1

---

### Meta-Review · Area_Chair1 · 2018-12-14

**Confidence:** 4
**Recommendation:** Reject

**Metareview:**

Pros:
- a method that obtains convergence results using a using time-dependent (not fixed or state-dependent) softmax temperature.

Cons:
- theoretical contribution is not very novel
- some theoretical results are dubious
- mismatch of Boltzmann updates and epsilon-greedy exploration
- the authors seem to have intended to upload a revised version of the paper, but unfortunately, they changed only title and abstract, not the pdf -- and consequently the reviewers did not change their scores.

The reviewers agree that the paper should be rejected in the submitted form.